# Drugging the Undruggable *Trypanosoma brucei* Monothiol Glutaredoxin 1

**DOI:** 10.3390/molecules28031276

**Published:** 2023-01-28

**Authors:** Annagiulia Favaro, Giovanni Bolcato, Marcelo A. Comini, Stefano Moro, Massimo Bellanda, Mattia Sturlese

**Affiliations:** 1Molecular Modeling Section (MMS), Department of Pharmaceutical and Pharmacological Sciences, University of Padova, via Marzolo 5, 35131 Padova, Italy; 2Laboratory Redox Biology of Trypanosomes, Institut Pasteur de Montevideo, Mataojo 2020, Montevideo 11400, Uruguay; 3Department of Chemical Sciences, University of Padova, via Marzolo 1, 35131 Padova, Italy; 4Consiglio Nazionale delle Ricerche, Institute of Biomolecular Chemistry, 35131 Padova, Italy

**Keywords:** glutaredoxin, trypanosomiasis, virtual screening, NMR

## Abstract

Trypanosoma brucei is a species of kinetoplastid causing sleeping sickness in humans and nagana in cows and horses. One of the peculiarities of this species of parasites is represented by their redox metabolism. One of the proteins involved in this redox machinery is the monothiol glutaredoxin 1 (1CGrx1) which is characterized by a unique disordered N-terminal extension exclusively conserved in trypanosomatids and other organisms. This region modulates the binding profile of the glutathione/trypanothione binding site, one of the functional regions of 1CGrx1. No endogenous ligands are known to bind this protein which does not present well-shaped binding sites, making it target particularly challenging to target. With the aim of targeting this peculiar system, we carried out two different screenings: (i) a fragment-based lead discovery campaign directed to the N-terminal as well as to the canonical binding site of 1CGrx1; (ii) a structure-based virtual screening directed to the 1CGrx1 canonical binding site. Here we report a small molecule that binds at the glutathione binding site in which the binding mode of the molecule was deeply investigated by Nuclear Magnetic Resonance (NMR). This compound represents an important step in the attempt to develop a novel strategy to interfere with the peculiar Trypanosoma Brucei redox system, making it possible to shed light on the perturbation of this biochemical machinery and eventually to novel therapeutic possibilities.

## 1. Introduction

Human African trypanosomiasis (HAT), also known as sleeping sickness, is an infectious disease caused by protozoan parasites belonging to the genus Trypanosoma. The disease is transmitted through the bite of an infected tsetse fly acting as a biological vector [1]. Human African trypanosomiasis threatens millions of people in sub-Saharan Africa, in which the deficient access to health services leads to an inadequate diagnosis and treatment of cases [2]. During acute infection, the trypanosomes invade the blood, lymph, and subcutaneous tissues. In humans, this haemo-lymphatic stage of the disease is characterized by joint pain, headaches, and fever. Upon parasite invasion of the central nervous system (the meningo-encephalic stage), the patients suffer from changes in behavior, confusion, lethargy, and finally coma and death [1,2]. Few drugs are available for treating HAT and most of them are old fashioned (i.e., Pentamidine, Suramin, Melarsoprol, Eflornithine, Nifurtimox), present severe side effects, and require complicated treatment management for underdeveloped countries [3]. A more recently developed oral drug, Fexinidazole, has reduced the limitation for drug administration but is mostly indicated for earlier stages of the disease [1,4]. In addition, several species of African trypanosomes can infect other mammals—in particular farm animals—which in underdeveloped countries often represent a food source and a work tool, particularly in rural villages [5]. In this scenario, the development of potent and easily administered drugs represents an important goal for fighting this neglected disease.

One of the peculiarities that distinguish Trypanosomes from the large majority of organisms is their peculiar thiol-dependent redox metabolism based on trypanothione (T(SH)_2_) and on the flavoenzyme trypanothione reductase. This contrasts with the mammalian redox systems that rely on the reducing power supply of glutathione (GSH) and glutathione reductase and thioredoxin and thioredoxin reductase [6,7]. Within the thioredoxin-fold superfamily, glutaredoxin (GRX) represents a major class of proteins involved in cellular redox homeostasis and iron–sulfur cluster metabolism using low molecular weight thiols, such as GSH, as a cofactor [8]. GRX are small ubiquitous proteins that can be classified into two classes based on their structural organization, active site motif, and function. Class I GRX is characterized by having a CXXS or monothiolic active site motif and for being involved in iron–sulfur trafficking/sensing in different cellular compartments. In contrast, class II GRX has a dithiolic active site (CXXC), can eventually bind an iron–sulfur cluster, and displays GSH-dependent oxidoreductase activity [9].

Trypanosomes encode for two dithiolic Grxs and three monothiolic Grxs (1CGrx) [10]. 1CGrx1 from *Trypanosoma brucei brucei* (*T.b.* 1CGrx1) is the best studied 1CGrx from trypanosomatids. The protein is localized in the parasite mitochondrion [11], where it fulfills a function indispensable for the survival of the pathogen in vitro and in vivo [12,13]. The role of 1CGrx1 has not been fully clarified, but yeast complementation experiments and biochemical assays have shown that the protein participates in iron–sulfur cluster biogenesis [12,13]. *T.b.* 1CGrx1 is a protein of 184 amino acids, in which the first 41 residues are necessary for its mitochondrial translocation. Even more interestingly, *T.b.* 1CGrx1 presents an intrinsically disordered N-terminal extension (residues 42–77), preceding the Grx domain (residues 78–184), and exclusively conserved in trypanosomatids and some plants, but absent in other organisms. We recently revealed the structure of the mature form of *T.b.* 1CGrx1 including its intrinsically disordered region (IDR), which modulates the accessibility to the binding pocket [13,14,15]. It should be also noted that complementation studies performed in Saccharomyces cerevisiae deprived of its mitochondrial Grx5 indicate that inserting *T.b.* 1CGrx1 achieves only a modest rescue of the phenotype. This observation suggests a peculiar specificity of the structural and or biochemical specificity of *T.b.* 1CGrx [13,16] compared to a closely related monothiol eukaryotic Grx. For these reasons, the mature form of *T.b.* 1CGrx1 represents an interesting molecular target for the development of a new class of specific inhibitors.

In this work, two of the most effective strategies for the development of novel binders were used to evaluate its druggability and to identify potential inhibitors. We first focus our attention on the unique N-terminal IDR through a Fragment-Based Lead Discovery (FBLD) campaign. FBLD is one of the most established techniques to develop compounds when no other ligands are known [17]. The identification of fragments that can bind the mature form of *T.b.* 1CGrx1 is very challenging because most of the protein cavities are exposed to the solvent. In addition, the intrinsic flexibility of the N-terminal region decreases the possibility of the formation of a stable and well-defined active site pocket [18]. Therefore, we used Nuclear Magnetic Resonance (NMR) as the primary method to identify protein ligands because it is one of the most reliable techniques to detect weak protein–ligand binding, as would be expected for molecules with low molecular weight [19]. A second screening strategy consisted of the in silico identification of ligands at the GSH and T(SH)_2_ binding site of the mature *T.b.* 1CGrx1; this site plays a major role in iron–sulfur cluster binding and transference to target proteins. The computational screening approach made use of the structural information available for *T.b.* 1CGrx1 and the detailed epitope of GSH bound to Grx4 in *E. coli* [14,20].

## 2. Results and Discussion

### 2.1. Fragment Library Screening by ^1^H-^15^N SOFAST-HMQC

With the aim of selectively targeting the N-terminal intrinsically-disordered domain of *T.b.* 1CGrx1, which makes 1CGrx1 unique among all the glutaredoxins [13], we decided to perform an FBDD by NMR approach for two principal reasons. First, FBDD represents the best strategy to identify small molecules that can bind to challenging targets, such as 1CGrx1; second, NMR is the gold-standard technique to detect weak protein–ligand interactions typical of a fragment. Moreover, NMR experiments are carried out in solution, where the protein preserves its dynamic and conformational flexibility; this aspect is particularly relevant since the N-terminal extension of the protein is a disordered region that could eventually assume a more structured conformation upon binding. A total of 783 fragments (the features of the library are described in Section 3.3 and graphically summarized in Appendix A) were screened in mixtures against ^15^N-labeled *T.b.* 1CGrx1 acquiring ^1^H-^15^N SOFAST-HMQC spectra in the presence and in the absence of a 10-fold molar excess of each fragment. We decided to use a protein-based approach as a preliminary screening strategy because *T.b.* 1CGrx1 is a small 15 kDa protein, with NMR signals that are well distributed in the two dimensions of the spectrum and limited overlapping or broadening signals. The excellent yield of ^15^N-labeled *T.b.* 1CGrx1 expression allowed us to screen the fragment library using this approach, which requires high quantities of isotope-labeled protein. Moreover, the fast pulsing technique SOFAST-HMQC significantly reduced the acquisition time, permitting us to acquire a single ^1^H-^15^N correlation experiment in a rapid way (i.e., 25 min). Unfortunately, the analysis of the ^1^H-^15^N SOFAST-HMQC spectra in the presence and in the absence of the fragment mixtures revealed that none of the small molecules screened bind to *T. b.* 1CGrx1. In fact, no evident changes in the chemical shift or intensity of the protein resonances are observed upon addition of the fragments, indicating that the chemical environment of the protein or its local electron density does not significantly change in the presence of the fragments, and hence they are classified as no binders. This experimental evidence suggests that the intrinsically-disordered N-terminal region of *T.b.* 1CGrx1 does not undergo conformational changes leading to the formation of transient defined ligand-binding cavities, thus resulting in target intractability. Furthermore, it should also be noted that, quite surprisingly, despite that a relatively large fragment library was screened, no binders were identified for the well-defined cavity which is able to bind low-molecular-weight thiols in glutaredoxins.

### 2.2. Structure-Based Virtual Screening Directed at the GSH/T(SH)_2_ Binding Site

Since the FBLD did not result in a satisfactory hit selection, we decided to address our attention to the canonical binding site of the mature form *T.b.* 1CGrx1 by setting a computational pipeline aimed at a virtual screening, taking advantage of the structural information available for glutaredoxins in complex with its natural binder, GSH, and the recently solved structure of *T.b.* 1CGrx1. First, the experimental binding mode of GSH in complex with monothiol Grx4 (PDB ID: 2WCI, Figure 1) [20] was carefully analyzed and compared with the *T.b.* 1CGrx1 (Appendix A) active site to identify the key contacts of GSH with the target protein. Next, a pharmacophoric model depicting these interactions was developed and used as first filter in the virtual screening campaign (Figure 1).

The structural analysis revealed that the major interactions between Grx4 and GSH involve: (i) charge–charge interactions of residues Arg59 and Lys22 with the glycyl carboxylate of GSH (patch B, Figure 1A); (ii) a salt bridge between Asp85 and the positively charged amine group of γ-Glutamate; and hydrogen bonds between (iii) the cysteine backbone of GSH with the Phe71 backbone (patch A, in Figure 1A) and (iv) the γ-Glutamate side-chain carboxylate with a Cys84 backbone (patch C in Figure 1A). Most of the residues engaged in these interactions are highly conserved in *T.b.* 1CGrx1 (Figure 1C). A minor difference was observed with respect to the orientation adopted by Arg59 (Arg133 in *T.b.* 1CGrx1), which in Grx4 is pointing toward the binding site and in *T.b.* 1CGrx1 toward the bulk (see Figure 1C).

However, this sidechain is particularly flexible and among the 20 deposited conformations several conformations were reported for *T.b.* 1CGrx1 (PDB ID: 2MXN); one of them can partially restore the conformation in Grx4. Reasonably, a similar orientation to Gxr4 could be restored upon GSH binding since the positively charged side chain of Arg133 could be attracted by the negatively charged glycine moiety.

Taking together, the pharmacophore model for *T.b.* 1CGrx1 binders considered the presence of: (i) a negatively charged group to favor interaction with Lys96 and Arg133 (Lys22 and Arg59, respectively, in Grx4); (ii) ligands able to establish two hydrogen bonds with Ile145 (Phe71 in Grx4); and (iii) an aromatic interaction with Tyr106 (Phe32 in Grx4). The last feature has been included with the aim to increase the affinity of the protein for the ligand. In fact, GSH has a low affinity for *T.b.* 1CGrx1 and π–π stacking between Tyr106 and the binder may guarantee stability to the ligand–protein complex. In Figure 1, panel D, the features of the pharmacophoric model are shown.

This pharmacophoric model was used to guide a docking-based virtual screening in which the posing procedure of the docking algorithm considered the matching of the pharmacophoric outlined above. The advantage of this procedure is the integration of the pharmacophoric filter with the optimization of the ligand conformation within the binding site. In addition, the obtained docking score for each entry is accurate in prioritizing the ligands since it estimates the quality of the interaction established, while the pharmacophore search alone only returns the match of the features. The application of this pipeline led to the screening of 3.2 million commercially available compounds. The eight top-scoring compounds were then purchased to proceed with the biophysical characterization of the binding to 1CGrx1 (the structure of the eight compounds is reported in Appendix A).

### 2.3. Biophysical Characterization of the Binding between T.b. 1CGrx1 and Hit Compounds by NMR

To assess the binding to *T.b.* 1CGrx1 of the six hit compounds originating from the virtual screening (compounds 2 and 7 were not tested due to limited availability), these molecules were subjected to both protein- and ligand-based NMR experiments to minimize false-positive or false-negative results. As a primary strategy, we decided to use the most robust NMR technique to detect protein –ligand interactions; ^1^H-^15^N SOFAST-HMQC experiments were acquired in the presence and in the absence of a 10-fold molar excess of each compound. These experiments revealed that compound 1 binds to *T.b.* 1CGrx1.

In fact, as shown in Figure 2, panel A, the intensity of the protein signals is significantly perturbed in the presence of compound 1, indicating that the chemical environment of the protein changed because of the formation of the protein –ligand complex. Moreover, the disappearance and the decrease in intensity of some protein signals in the presence of compound 1 suggests that the system is in an intermediate exchange on the chemical shift timescale, which makes the compound 1 dissociation constant (*K*_D_) very difficult to derive with a 2D titration. On the contrary, the analysis of the variations in protein signal intensities due to the ligand binding allowed us to map the binding site, which was defined by those residues experiencing a change in peak intensity higher than two standard deviations (2ơ) from the average. As shown in Figure 2, panels B and C, the most perturbed residues are Val92, Thr93, Ile95, Lys96, Val126, Val131, Val132, Glu138, Glu141, and Val155, which correspond to residues nearby the putative GSH binding pocket. Interestingly, the three residues of the IDR that are significantly perturbed (Ile54, Phe61, and Ala76) belong to regions of the disordered tail proposed to transiently interact with the pocket [15]. This experimental evidence indicates that the ligand binding is specific and nicely agrees with the computational approach, where the ligand was rationally designed into the GSH binding site.

To further validate the binding of compound 1 to *T.b.* 1CGrx1 we used two of the most common ligand-based NMR techniques to study protein –ligand interactions: WaterLOGSY and STD experiments were acquired in the presence and in the absence of the protein. These experiments clearly confirm the binding of compound 1 to *T.b.* 1CGrx1. In fact, the WaterLOGSY experiment in the presence of the protein (Figure 3, panel A, in blue) reveals that the signals of the ligand assume the same negative sign as the protein, indicating that the ligand acquired the protein magnetization as intermolecular Nuclear Overhauser Effect (NOE) due to the formation of the protein –ligand complex. Otherwise, the signals arising from DTT, DMSO, and the internal standard DSS have opposite phases (positive), suggesting that they do not interact with *T.b.* 1CGrx1. Moreover, the proton signals of the two phenyl groups of the aromatic amide clearly undergo a strong sign inversion, indicating that this portion of the molecule is in close contact with the protein, while the signal of the unshielded hydrogen of the heterocyclic moiety disappears, evincing that this position is solvent exposed. The epitope mapping by NMR agrees with the predicted pose originated from the computational studies (Figure 1, panel F), which show that the two phenyl groups of the ligand are oriented toward the binding pocket, where they form π–π stacking interactions. The two rings have a different accessibility to the solvent; one of them in deeply inserts toward the protein surface, but unfortunately, the symmetry leads to the averaging of the signal intensity of the two signal groups originating from the two slightly different two-ring orientation. The STD experiments returned the same results, thus uniquely confirming the binding of compound 1 to *T.b.* 1CGrx1 and its orientation upon binding. In fact, as shown in Figure 3, panel B (in turquoise-green) all the signals of the ligand appear, indicating that in the on-resonance experiment, the ligand acquired the saturated protein magnetization via intermolecular spin diffusion because of compound 1 binding to *T.b.* 1CGrx1. Moreover, the intensity of the proton signals of the two phenyl groups is higher (STD factor (%) = 100, Figure 3, panel B, red columns) due to a more efficient magnetization transfer between the protein and the ligand, proving the proximity of this ligand moiety to the protein upon binding.

## 3. Materials and Methods

### 3.1. Computational Studies

#### 3.1.1. Molecular Modeling

The database used in the virtual screening was the Molport database of commercially available compounds (https://www.molport.com/; accessed on 4 December 2022). This virtual database of 3.2 million compounds was prepared using OpenEye [21] suite for the following steps: partial charge calculation, prediction of the most stable tautomer, and prediction of the most probable protonation state at pH 7.4. An initial 3D conformation for each molecule was calculated using Corina classic [22].

The two protein structures (PDB codes 2WCI and 2MXN) [14,20] used in this study were prepared using MOE [23]. In this preparation step, water molecules were removed as well as other co-solvent molecules and ions. The tool Protonate3D was used to calculate the protonation state of each residue at pH 7.4.

#### 3.1.2. Pharmacophore Modeling and Molecular Docking

The protein used for docking was 1CGrx1 from *Trypanosoma brucei* (PDB code 2MXN). The binding site was identified by superimposing the structure of holo-Grx4 from *Escherichia coli* (PDB code 2WCI) where glutathione is present coordinating a [Fe2S2] cluster. The conserved GSH site of 1CGrx1 was used for the docking calculation. The Pharmacophore model used in the docking search was prepared using MOE [23]. A detailed description of this model is reported in the results section. The virtual screening was performed by a molecular docking protocol using MOE with the following settings: 100 poses were used for placement and 10 for refinement, the Pharmacophore method was used for placement (The generated poses must fit the pharmacophore model), and the London dG scoring function was used for a first ranking procedure while the GBVI/WSA dG scoring was used for the refinement and the final sorting.

### 3.2. Recombinant T.b. 1CGrx1 Expression and Purification

The mature form of *T.b.* 1CGrx1 lacking the mitochondrial targeting sequence (∆1-41) was expressed as an N-terminal fusion protein to a 6xHis *E. Coli* Thioredoxin (Trx) 1 tag, linked to a 3C-type Tobacco Etch Virus (TEV) protease recognition sequence. Specifically, the coding region of the mature form of the *T.b*. 1CGrx1 was inserted into the vector pET-trx1b (kindly provided by Günther Stier, EMBL-Heidelberg), and the expression plasmid was transformed into the *E. coli* strain BL21 (DE3) using the thermal shock method. ^15^N uniformly labeled protein was produced in M9 minimal medium enriched with ^15^NH_4_Cl (1 g/l) as the sole nitrogen source. Kanamycin (50 µg/mL) was used as a selection agent. The transformed bacteria were grown at 37 °C and 180 rpm to an OD600 of 0.6–0.8, and the recombinant protein expression was then induced overnight at 23 °C and 180 rpm with 200 µM isopropyl β-D-thiogalactopyranoside (IPTG). The cells were harvested by centrifugation (5000 rpm, 25 min, 4 °C) and the cell pellet was resuspended in 50 mM sodium phosphate, pH 8.0, 300 mM NaCl, 10 mM β-mercaptoethanol (βME), and 15 mM imidazole (buffer A) containing the serine protease inhibitor phenylmethylsulfonyl fluoride (PMSF) (1 mM) and the cysteine protease inhibitor N-[N-(L-3-trans-carboxyoxiran-2-carbonyl)-L-leucyl]agmatine (E-64) (10 µM). The cells were lysed by sonication and the lysate was clarified by centrifugation (14,000 rpm, 45 min, 4 °C). The recombinant tag-free *T.b.* 1CGrx1 was then purified as described by Manta et al. [13]. Briefly, the cell lysate was loaded into a HisTrap^TM^ column (Cytiva, Washington, DC, USA) previously equilibrated with buffer A. The His-tagged fusion protein was eluted using an imidazole linear gradient (from 15 to 500 mM), and subsequently, it was buffer exchanged into 50 mM sodium phosphate, pH 8.0, 300 mM NaCl, and 10 mM βME on a HiPrep 26/10 desalting column (Cytiva). To remove the N-terminal 6xHis-Trx1 tag the recombinant fusion protein was incubated overnight at 4 °C with TEV protease. A further Ni^2+^-affinity chromatography was performed to separate the tag-free *T.b.* 1CGrx1 from the His-tagged protease and Trx1. Finally, the *T.b.* 1CGrx1 was purified by Size Exclusion Chromatography (SEC) on a HiLoad™ 16/60 Superdex™ 75 prep-grade column (Cytiva, Washington, DC, USA), equilibrated with 50 mM sodium phosphate, pH 7.1, 150 mM NaCl, 2 mM dithiothreitol (DTT), and 0.04% (*w*/*v*) NaN_3_ (buffer B).

### 3.3. Fragment Library Preparation and Characterization

Our fragment library is composed of 783 molecules, which were selected according to the modern trends in fragment library design: (i) on average they are Rule Of 3 (RO3) compliant (94 Da ≥ Molecular Weight (MW) ≤ 300 Da, average = 187 Da; −4 ≥ logP ≤ 4, average = 1.4; 0 Å^2^ ≥ Topological Surface Area (TPSA) ≤ 118 Å^2^; average = 44 Å^2^; 0 ≥ H-bond acceptors ≤ 6, average = 2; 0 ≥ H-bond donors ≤ 5, average = 1; 0 ≥ rotatable bonds ≤ 10, average = 2; 7 ≥ number of Heavy Atoms (HA) ≤ 22, average = 13); (ii) their solubility is proved in dimethyl sulfoxide (DMSO) (100 mM) and in aqueous solution (1 mM); (iii) most of them contain scaffolds, which are commonly present in drug molecules and are 3D-shaped (0 ≥ Normalized Principal Moment of Inertia 1 (npr1) ≤ 0.6, average = 0.2; 0.5 ≥ npr2 ≤ 1, average = 0.8). An overview of the calculated properties of the fragment library is reported in Appendix A. Moreover, our fragment library does not contain PAINS (Pan Assay INterference compoundS) molecules [24,25], which could generate false positives since they can lead to the formation of covalent bonds, chelating species, or aggregates. Specifically, 192 fragments were purchased from Otava, 243 from Key Organics, and 111 from Life Chemicals, while 237 fragments were kindly provided by Graziano Lolli, University of Trento.

The fragment library was prepared and characterized according to the following procedure: each powder was solubilized in deuterated DMSO at 100 mM and stock solutions were plated in 96-well plates and stored at −20 °C. To assess the solubility in an aqueous environment, stock solutions were diluted at 1 mM in 20 mM sodium phosphate, pH 7.0, and 10% v/v deuterated water to acquire ^1^H NMR spectra. Then, 4,4-dimethyl-4-silapentane-1-sulfonic acid (DSS) was added to each NMR tube (100 µM) as an internal standard.

The NMR spectra were processed with Topspin 4.1.3 (Bruker BioSpin GmbH, Rheinstetten, Germany) software and the assignment was made using the CMC-assist tool of Topspin 4.1.3 (Bruker BioSpin GmbH, Rheinstetten, Germany), followed by manual control of each spectrum. The fragment concentration was calculated using the following equation:(1)$$C_{f}=\frac{I_{f}}{I_{d}}\cdot \frac{N_{d}}{N_{f}}\cdot C_{d}$$
where C, I, and N are the concentration (mol/L), the integral area, and the number of protons, respectively, giving rise to the signal of the fragment of interest (f) and the internal standard DSS (d).

### 3.4. NMR-Based Approaches

All the NMR experiments were acquired at 298 K on a Bruker 600 MHz spectrometer equipped with an autosampler (69 samples), and a nitrogen-cooled cryoprobe.

#### 3.4.1. NMR-Based Fragment Screening

The fragment library was screened in mixtures of 10-fold molar excess (350 µM) against ^15^N-labeled mature *T.b.* 1CGrx1 (35 µM) with a protein-based approach performing ^1^H-^15^N Heteronuclear Multiple Quantum Coherence (HMQC) coupled to SOFAST pulse scheme [26] experiments. The SOFAST-HMQC experiments were acquired with 48 scans, a recovery delay of 100 ms before each scan, and 162 increments in the indirect dimension for a total acquisition time of around 25 min. The number of points in the indirect dimension was doubled by linear prediction in the processing scheme. The samples were prepared as follows: the protein was diluted at 35 µM into buffer B and plated into 2 mL 96-well plates (Sigma Aldrich) and deuterated water (5% *v*/*v*) was added to each well. The fragments were then plated using the robotic liquid handler OT-2 (Opentrons), and the pH of all NMR samples was carefully adjusted at 7.10 ± 0.05 before each measurement to avoid any chemical shift perturbations (CSPs) induced by the pH. The screening was performed by adding to the protein 7 fragments, for a total of 112 mixtures. Control wells included protein incubated in the absence of fragment mixtures. The percentage of deuterated DMSO in every sample was 2.5%, which does not induce significant CSPs on *T.b.* 1CGrx1.

The NMR spectra were processed and analyzed by Topspin 4.3.1 (Bruker BioSpin GmbH, Rheinstetten, Germany) and Sparky [27]. The analysis of the ^1^H-^15^N SOFAST-HMQC spectra was performed using an in-house developed program which, from the peak list of the apo-protein and of the protein in the presence of a fragment mixture, automatically quantifies the average value of all the CSPs according to the following equation:(2)ΔδNH=δ1HMix−δ1HAP02+δ15NMix−δ15NAPO25212
where δ^1^H_APO_ and δ^15^N_APO_ represent the chemical shifts of the apo-protein, and δ^1^H_MIX_ and δ^15^N_MIX_ represent the chemical shifts of the protein in the presence of the fragment mixture [28]. The peak list of apo 1CGrx1 was derived from the *T.b.* 1CGrx1 backbone assignment reported in the paper of Sturlese et al. [29] (BMRB Entry: 19736). To identify the mixtures containing possible binders, we calculated the average value of CSPs of the five most perturbed residues (TOP 5 averaged CSPs); we should have selected the six mixtures (5% of the total mixtures) which showed the highest value of TOP 5 averaged CSP. We decided to use this cutoff since the hit rates for fragment-based screenings are typically 3–5% [30].

#### 3.4.2. NMR Binding Analysis of the Virtual Screened Compounds

The eight molecules that originated from the virtual screening were purchased from Molport (www.molport.com; accessed on 4 December 2022) and six of them were tested against mature *T.b.* 1CGrx1 using both protein- and ligand-based NMR approaches. Compounds 2 and 7 (Appendix A), were not tested due to their limited availability. As a primary strategy, ^1^H-^15^N SOFAST-HMQC spectra were acquired on ^15^N-labeled *T.b.* 1CGrx1 (10 µM) in the absence or presence of a 10-fold molar excess (100 µM) of the compound. The protein was diluted at 10 µM into buffer B, and deuterated water (5% *v*/*v*) and DSS (50 µM) were added to each NMR tube prior to the addition of the compounds. The pH of all NMR samples was carefully adjusted at 7.10 ± 0.05 before each measurement to avoid any CSPs induced by the pH. The percentage of deuterated DMSO in every sample was 0.5%, which does not induce significant CSPs on *T.b.* 1CGrx1. The SOFAST-HMQC experiments were acquired with 180 scans, a recovery delay of 100 ms before each scan, and 162 increments in the indirect dimension for a total acquisition time of around 2 h.

The 2D-NMR spectra were processed and analyzed by Topspin 4.3.1 (Bruker BioSpin GmbH, Rheinstetten, Germany). Specifically, we quantified the peak intensities of the apo-protein and of the protein in the presence of the compound. Peak intensities of the apo- and holo-protein were then normalized on Gly48 peak intensity, which does not change during the addition of the compounds. Subsequently, the percentage ratio between the peak intensities of the holo- and apo-form of each residue (R_x_) was calculated according to the following equation:(3)$$R_{x}=\frac{I_{HOLO_{x}}}{I_{APO_{x}}}\times 100$$
where I_HOLOx_ and I_APOx_ represent the normalized peak intensity of the residue x in the ligand and apo-form, respectively. The change in peak intensity (∆I) of each residue was then calculated as the difference between the percentage ratio of Gly48 (that corresponds to 100%) and the percentage ratio of each residue; we used this parameter to identify *T.b.* 1CGrx1-binding molecules and their binding site. To identify the binding site, we calculated the standard deviation (ơ) of the change in peak intensities (∆I) of all protein residues, and 2ơ was used as the threshold cutoff to identify only true positives (less sensitive but more specific).

Ligand-based NMR experiments were used to confirm the binding of the molecule selected from the protein-based approach. The ligand-based procedure consisted of Saturation Transferred Difference (STD) [31] and Water-Ligand Observed via Gradient SpectroscopY (WaterLOGSY) [32] experiments in the absence and presence of the protein. The NMR samples were prepared in buffer B and consisted of 100 µM of compound in 0.4% deuterated DMSO, 5% (*v*/*v*) deuterated water, 50 µM DSS, and 5 µM *T.b.* 1CGrx1. The WaterLOGSY experiments were acquired with 232 scans, a recovery delay of 4.5 s before each scan, and a mixing time of 1.5 s. They were performed with a 180° inversion pulse applied to the water signal at 4.7 ppm using a Gaussian-shaped selective pulse of 7.5 ms. STD experiments were acquired with 512 scans, and a recovery delay of 2 s before each scan. Selective saturation of the protein at 0.4 ppm was achieved by a 2.5 ms pulse train (60 Gaussian pulses of 50 ms separated by an inter-pulse delay of 1 ms) included in the relaxation delay, and a 25 ms spin lock was used to remove the broad residual protein signal. In both experiments, water suppression was achieved by the excitation sculpting pulse scheme [33]. STD experiments also allowed us to calculate the STD factor (STD_f_) for the different groups of protons of the ligand, which was calculated using the following equation:(4)$$STD_{f}=\frac{I_{\left(+p\right)}-I_{\left(-p\right)}}{I_{0}}$$
where I_(+p)_, and I_(−p)_ are the peak integral in the STD spectrum in the presence and absence of the protein, respectively, while I_0_ is the peak integral of the corresponding peak in the off-resonance experiment in the presence of the protein.

These experiments were processed and analyzed by Topspin 4.3.1 (Bruker BioSpin GmbH, Rheinstetten, Germany).

## 4. Conclusions

Human and animal trypanosomiasis caused by parasites from *T. brucei* complex are widespread in sub-Saharan Africa; however, few therapeutic agents are available for these diseases. With the aim of identifying drug-like molecules against a novel and indispensable protein for the pathogen, we performed a drug discovery campaign to target the mature form of *T.b.* 1CGrx1 by combining fragment-based and virtual screening approaches. Only the in silico approach was capable of originating a validated hit. It should be considered that the hit rate for fragment-based spans between 0.1–10% and more often between 3–5% depending on the druggability of the target [30,34]. Considering the poor outcomes in our fragment screening (no binders detected out of 783 fragments screened), *T.b.* 1CGrx1 can be considered a protein with low druggability. In line with *T.b.* 1CGrx1 function in Fe/S trafficking/sensing, the natural ligands (GSH and T(SH)_2_) bind with very low affinity to the protein active site—in the order of two digits millimolar—suggesting that the development of a strong binder is challenging. The only hit compound we identified originated from a virtual screening, and its binding to the target protein was experimentally confirmed by NMR. The NMR data proved that compound 1 binds to the GSH binding site. Ligand-based experiments also provided information on the epitope mapping of the ligand: the two phenyl groups of the aromatic amides are oriented toward the binding pocket and the unique aromatic proton of the heterocyclic moiety is solvent exposed. Worth noting, the epitope mapping is in full agreement with the binding mode obtained by molecular docking. These experimental findings represent the starting point for the rational hit-to-lead optimization of compound 1 that should address the improvement of water solubility and target affinity, and further testing of its biological activity and selectivity (e.g., other GRX). In this regard, the chemical modification of the two phenyls may offer the possibility of establishing a further interaction with Leu158 and Asp159, two residues that in *E. coli* Grx4 stabilize the GSH conformation via a network of hydrogen bonds with their backbone and the Asp159 sidechain. To our knowledge this represents the first report of a rational drug discovery campaign against a Class I (monothiol) glutaredoxin.

## Figures and Tables

**Figure 1 molecules-28-01276-f001:**
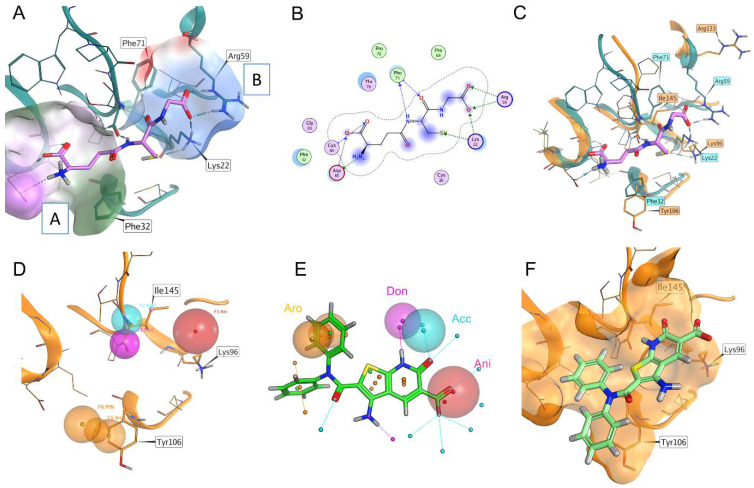
Panel (**A**): Experimental binding pose of GSH in Grx4 (From *Escherichia coli*). Surface patch A is colored according to lipophilic potential, where green is more lipophilic and purple is more hydrophilic. Surface patch B is colored according to electrostatic potential, where blue is positively charged and red negatively charged. Panel (**B**): Schematic representation of the experimental binding mode reported in Panel (**A**). Blue arrows are hydrogen bonds, green arrows are charge–charge interactions, blue highlights on the ligand’s atoms are exposed solvent. Panel (**C**): Superposition of the glutathione binding site in 1CGrx1 (orange) and Grx4 (light blue); the experimental binding pose of glutathione in Grx4 is also reported (purple). Panel (**D**): The pharmacophore model. Red sphere: negatively charged group. Purple sphere: hydrogen bond donor with the backbone of Ile145. Light blue sphere: hydrogen bond acceptor with the backbone of Ile145. Orange spheres: position and orientation in space of aromatic groups, forming a π–π stacking with the side chain of Tyr106. Panel (**E**): The superimposition of the pharmacophoric model with compound 1. Panel (**F**): Pose of compound 1, *T.b.* 1CGrx1; the solvent-accessible surface is represented in orange.

**Figure 2 molecules-28-01276-f002:**
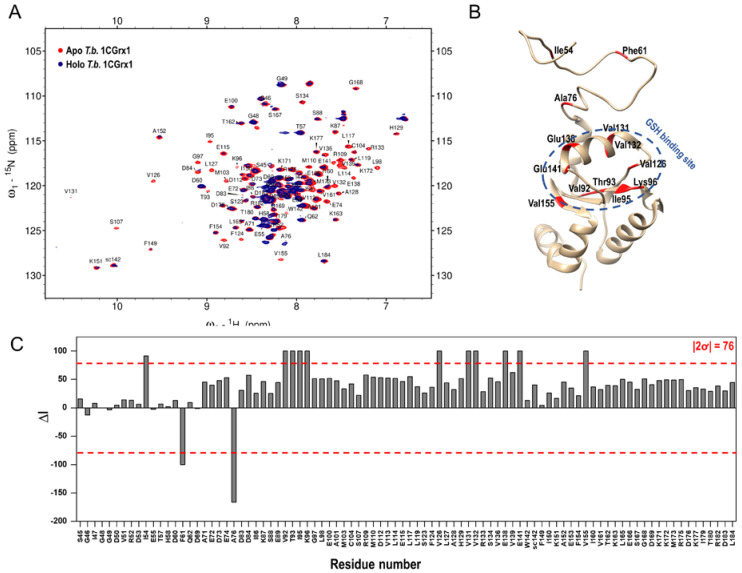
Experimental confirmation of compound 1 binding to T.b. 1CGrx1 by protein-based NMR techniques. (**A**) Superposition of ^1^H-^15^N SOFAST-HMQC spectra in the presence (blue) and in the absence (red) of a 10-fold molar excess of compound 1. (**B**) Peak intensity perturbations induced by A-4 binding. The change in peak intensity (∆I) was calculated as described under “Materials and Methods”. The red dashed line represents our threshold cutoff (2ơ). (**C**) Binding site definition by peak intensity perturbation analysis: the most perturbed residues were mapped in red in the three-dimensional structure of T.b. 1CGrx1 (PDB ID: 2MXN).

**Figure 3 molecules-28-01276-f003:**
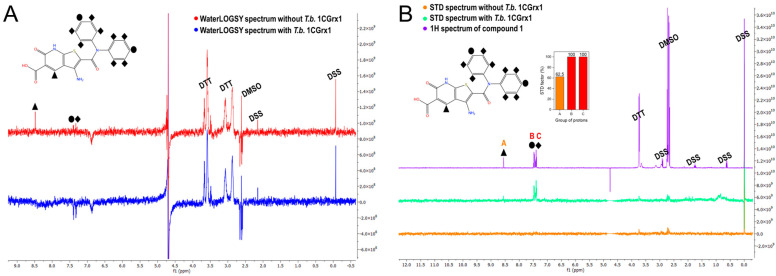
Experimental confirmation of compound 1 binding to T.b. 1CGrx1 by ligand-based NMR techniques. (**A**) WaterLOGSY spectra of compound 1 (100 µM) in the presence (blue) and in the absence (red) of T.b. 1CGrx1 (5 µM). (**B**) STD spectra of compound 1 (100 µM) in the presence (green) and in the absence (orange) of T.b. 1CGrx1 (5 µM). The assignment of the signals is reported on the 2D structure of the molecule. (**B**) STD spectra of compound 1 (100 µM) in the presence (turquoise-green) and in the absence (orange) of T.b. 1CGrx1 (5 µM), and 1H NMR spectrum of compound 1 (100 µM) in 50 mM sodium phosphate, pH 7.1, 150 mM NaCl, 2 mM DTT, and 0.04% (*w*/*v*) NaN3, and STD binding epitope mapping of compound 1. The STD factor (%) for each group of protons was calculated as reported in “Materials and Methods” and normalized to the most intense one (groups B and C of protons). The assignment of the signals is reported on the 2D structure of the molecule.

## Data Availability

Not applicable.

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
