# Peer review of "Drugging the Undruggable Trypanosoma brucei Monothiol Glutaredoxin 1"

_molecules, 2023, doi:10.3390/molecules28031276_

Round 1

Reviewer 1 Report

The manuscript entitled "Drugging the Undruggable Trypanosoma brucei Monothiol Glutaredoxin 1 " by Favaro and co-workers describes the identification of a hit compound to target 1CGrx1 combining fragment based and virtual screening approaches. The manuscript is overall properly conducted and written and it is suitable for the readers of the Molecules. However, I would like to underline that the quality of the images in the manuscript, is very bad in terms of resolution. It was barely possible to see the images and evaluate the data shown. This issue should be fixed before publication.

Additional comments:

1. What is the main question addressed by the research? 

The authors were able to find a hit compound for a particularly challenging protein, namely the Monothiol Glutaredoxin 1. 

2. What does it add to the subject area compared with other published material? 

The topic is certainly original as it provides a compound that could represent a starting point in the search for a strategy to interfere with the T. brucei redox system. 

3. What specific improvements should the authors consider regarding the methodology? What further controls should be considered? 

I believe that the article is suitable for publication and no further controls are needed. 

4. Are the conclusions consistent with the evidence and arguments presented and do they address the main question posed?

Conclusions are consistent with the data presented.

5. Are the references appropriate? 

References appear appropriate 

6. Please include any additional comments on the tables and figures. 

The resolution of the figures within the manuscript is poor, I believe that the authors had problems in uploading the figures. While they should be ameliorated (only in terms of resolution) before publication (especially figure 2), it was however possible to evaluate the data presented in them.  

Author Response

We warmly thank the reviewer for the time spent reviewing our manuscript. 
The reviewer is entirely right about the quality of the figures that for some "technical reason" appear at low resolution in the generated PDF file while in the original manuscript they are at 300 DPI. We will certainly take care of this aspect n the resubmission and during the proofreading stage. 

Reviewer 2 Report

The manuscrit by Favaro et al. is an example of combination of NMR and virtual screening for de novo identification of small molecule binders for proteins. The science is robust and the results appear convincing. The authors can improve the manuscript as suggested : 

- the pharmacophore used for the fragment screening should be displayed in the absence of the compound in Figure 1D, and in another panel, the phamacophore could be superimposed to the chemical structure of the hit. 

- in Figure 3 panel B the STD spectrum seems to require phasing.

- in Figure 3 panel B the 1H normal spectrum should be presented. The authors could calculate the STDfactor per 1H for the 1H of the phenyl and compare to the STD factor of the 1H of the heterocyclic moiety. 

- the authors should provide a picture of the docking pose where the accessibility of the 1H would be easier to observe (for example with compound 1 bound to CGRX1 shown in surface mode)

- it could be interesting to test fragments resulting from the deconstruction of compound 1 using NMR-2D at higher concentrations (1 mM); also changing the NMR field and temperature might be useful. In general fragment screening might be performed at higher concentrations as the protein is poorly druggable.

- as the compound does not interact with the IDR, the authors might modify the introduction to focus on the difference in the active sites between the TB GRX as compared to other GRX.

Author Response

We kindly thank the reviewer for her/his time and for the valuable suggestions.  The answers to reviewer comments are here commented point-by-point.

Q: the pharmacophore used for the fragment screening should be displayed in the absence of the compound in Figure 1D, and in another panel, the phamacophore could be superimposed to the chemical structure of the hit. 

A: Figure 1 was modified following the indication of the reviewer including two new panels

Q: in Figure 3 panel B the STD spectrum seems to require phasing.

A: Figure 3B was modified accordingly. 

Q: in Figure 3 panel B the 1H normal spectrum should be presented. The authors could calculate the STDfactor per 1H for the 1H of the phenyl and compare to the STD factor of the 1H of the heterocyclic moiety. 

A: Figure 3B now includes also the 1H spectrum

Q: the authors should provide a picture of the docking pose where the accessibility of the 1H would be easier to observe (for example with compound 1 bound to CGRX1 shown in surface mode)

A: Figure 1 was modified following the indication of the reviewer including two new panels

Q: it could be interesting to test fragments resulting from the deconstruction of compound 1 using NMR-2D at higher concentrations (1 mM); also changing the NMR field and temperature might be useful. In general fragment screening might be performed at higher concentrations as the protein is poorly druggable.

A: We thank the reviewer for the brilliant observation. Actually, we will face the optimization of the hit and there are several successful examples in which the adoption of this strategy was successfully used in the past.  We certainly consider this suggestion in the future Hit2Lead stage.  

Q: as the compound does not interact with the IDR, the authors might modify the introduction to focus on the difference in the active sites between the TB GRX as compared to other GRX.

A: The introduction now integrates this aspect.  

Reviewer 3 Report

Favaro et al. report the discovery of a small molecule that may be the prototype drug for Trypanosoma brucei. The article is well written, logical and consistent. The choice of methods for solving the set goal is justified and adequate. There is some inconsistency in the definition of the dissociation constant. The authors mention that the compound 1 dissociation constant (KD) very difficult to derive with a 2D NMR titration experiments (line 210). But KD can be obtained from WaterLOGSY experiments. WaterLOGSY method is effective for measuring KD in the millimolar range. But on the other hand, perhaps the authors consider this issue not so important for solving the problem at this stage. There is only one minor remark.

 1. The target for the drug was the monothiol glutaredoxin 1 (T.b. 1CGrx1), consisting of 184 amino acid residues (line 72). All NMR experiments were performed with a 15 kDa recombinant protein (line 113, paragraph 3.2 and Fig. 2), which is a truncated version of the T.b. 1CGrx1. But in the text for this truncated recombinant protein uses exactly the same name as for the full wild protein T.b. 1CGrx1, which introduces some confusion. I suggest using different names for recombinant protein and wild protein.

Despite the existing comment, I recommend the article for publication.

Author Response

We thank the reviewer for the time spent reviewing our manuscript and for the valuable comments. 

Q:  The authors mention that the compound 1 dissociation constant (KD) very difficult to derive with a 2D NMR titration experiments (line 210). But KD can be obtained from WaterLOGSY experiments. WaterLOGSY method is effective for measuring KD in the millimolar range. But on the other hand, perhaps the authors consider this issue not so important for solving the problem at this stage. There is only one minor remark.

The reviewer is right and actually, we tried to estimate the KD of compound 1 by a waterLOGSY titration. However, the poor solubility of the ligand in aqueous solution (100 µM) does not allow the protein saturation. We have tried different methods, including fluorescence since the unique TRP is closely located to the GSH binding site but, unfortunately the compound 1 itself presents a strong fluorescence leading to two contemporary inner filter effects. Preliminary experiments to overcome those effects suggested a very low micromolar affinity range. However, the strong fluorescence of the ligand limited a very accurate estimation.  However, all the experiments performed indicate a binding event suggesting this hit as a candidate for future optimization toward a lead compound. Our priority in the near future will be to improve the solubility and affinity. In this scenario, it will easier to characterize the lead binding.

Q: The target for the drug was the monothiol glutaredoxin 1 (T.b. 1CGrx1), consisting of 184 amino acid residues (line 72). All NMR experiments were performed with a 15 kDa recombinant protein (line 113, paragraph 3.2 and Fig. 2), which is a truncated version of the T.b. 1CGrx1. But in the text for this truncated recombinant protein uses exactly the same name as for the full wild protein T.b. 1CGrx1, which introduces some confusion. I suggest using different names for recombinant protein and wild protein.

All the experiments in this manuscript were performed using the mature form of T.b. 1CGrx1, The mitochondrial targeting sequence (1-41) was not included, to better represent the functional form of the protein.  In the past, we solved the structure and characterized the binding to GSH/T(SH)2 with the globular domain Δ76 but since the binding profile was different from the mature form we decide to focus on this form. To avoid misunderstanding, we clarified in the manuscript that we performed the experiments on the mature 1CGrx1.